# The Sustainability of a Community of Inquiry in Online Course Satisfaction in Virtual Learning Environments in Higher Education

**M. Khalid M. Nasir** [1,*,†] ⬤ **and Abdul Hafaz Ngah** [2,†] ⬤

1   Faculty of Education, Universiti Kebangsaan Malaysia, Bangi 43600, Selangor, Malaysia
2   Faculty of Business, Economics & Social Development, Universiti Malaysia Terengganu, Kuala Terengganu 21030, Terengganu, Malaysia
*   Correspondence: mdkhalid@ukm.edu.my
†   These authors contributed equally to this work.

**Abstract:** Teaching and learning online is quite challenging. Both require an additional capacity and effort to withstand ongoing engagement in a virtual learning environment. Nonetheless, there have been cases of dissatisfaction with virtual learning environments due to the lack of engagement and poor interaction between the instructor, students, and content, which may affect how students learn online. This study presents a cross-sectional survey that was designed to re-examine the theoretical model of the Community of Inquiry (CoI), and to examine the structure of course satisfaction using SmartPLS 3.3.8 for multivariate statistical analysis. The CoI and the course satisfaction instruments were adapted in this study. The reflections of the CoI are then assumed to form type II second-order constructs to determine their effect on student satisfaction with the course. The findings revealed that teaching, social, and cognitive presence in the CoI have a significant influence on students' satisfaction with the courses that they are enrolled in. These results provide a direction for further research on the CoI in online learning by extending a framework that incorporates online learners as one of the essential stakeholders in education. Therefore, the results presented here are only applicable to certain courses, and it would be meaningful to investigate academic achievement and motivation, and to compare them between specific courses or subjects to find out which courses have lower or higher levels of presence.

**Keywords:** teaching presence; social presence; cognitive presence; Community of Inquiry (CoI); course satisfaction; Open Distance Learning (ODL)

## 1. Introduction

Online learning has grown rapidly worldwide due to market demand and student needs. The need for online learning normally comes from people who are unable to physically attend classes and meet their teachers or lecturers face-to-face. As such, online learning allows students to pursue their studies via online courses or programs. However, the information on the effectiveness of online learning remains sketchy, and there has been very limited information available in the literature. There have been a number of concerned issues; notably, self-directed learning, assessment tools, immediate teacher feedback, teacher expertise and personal traits, technical problems, poor engagement, and a lack of interaction, which trigger dissatisfaction among online students [1]. Apart from that, other issues, such as limited access to hardware and software, personal circumstances, lack of experience [2], limited feedback, unsupervised online learning [3], a lack of motivation, technology and Internet issues, and data privacy and security concerns [4], also affect satisfaction [2] and contribute to the failure of online courses, which leads to increasing dropout rates [5–7]. These are a number of crucial issues that need to be taken into consideration in order to sustain the quality of online courses.

Other concerned issues are in regard to the adult learners' characteristics (e.g., age, gender, educational background, experience); external factors (e.g., carrier, family, time); and internal factors (e.g., self-regulated learning, learning strategies, goals, digital skills, technology acceptance), as found by Lu et al.'s systematic review [8]. These issues demand solutions that take the adult learners' needs into consideration. In other words, teachers, peers, and the content need to be effectively incorporated in the virtual learning environment with more engagement and flexibility [9,10]. One popular model that measures these concepts is known as the Community of Inquiry (CoI) model, where engagement and interaction, also known as presence, are adapted.

As for the uniqueness of this study, firstly, it focuses on student satisfaction toward online courses; particularly, satisfaction with the course goals (a course's online learning objectives), course content (syllabus/pro forma), and course discussion. Secondly, this study contributes to the literature on the CoI. Based on researchers' search of Clarivate and Web of Science databases from January 2018 to July 2022, there have been a very limited number of studies on: (1) online learning applying PLS analysis, and (2) type II second-order constructs to seek CoI effects on student satisfaction.,

Therefore, this paper intends to re-examine the theoretical CoI model, focusing on satisfaction with online education courses in Malaysia. It is also expected to present a meaningful framework to the instructors of online courses in order to enhance teaching, and the social and cognitive presence in their teaching and learning activities online in international contexts. Presence, as acknowledged by scholars in the field of online learning, is an essential element to consider when offering online courses in the CoI framework. This paper is guided by three main research questions: RQ1: Does teaching presence play a role in improving course satisfaction? RQ2: Does social presence play a role in improving course satisfaction? RQ3: Does cognitive presence play a role in improving course satisfaction? This study was designed in such a way that it could answer several hypotheses simultaneously, and this is elaborated in the hypotheses section.

## 2. Related Past Study

### 2.1. Challenges and the Current Practice of Online Learning

A large number of higher education institutions offer online courses. In 2015, over seven million students were enrolled in at least one online course [11] in the United States. In addition, the number of students registering online courses worldwide was increasing drastically in 2021: nearly 92 million students from North and Latin America, 189 million from Europe and the Middle East, and 2.2 million from Asia Pacific and Africa [12]. On one hand, there has been an increase of students registered in online courses, especially during the pandemic, but on the other hand, there has been a steady trend towards attrition. This scenario has puzzled many educators. Furthermore, many online courses are not sustainable and are not able to survive [1,8,11].

Isolation, boredom, and dropping out of courses [6,7], as well as student dissatisfaction, are among the issues experienced in online learning [13]. Another essential issue in online learning is peer engagement, with cognitive activities emphasizing student interaction with their online instructors representing one of the most significant aspects requiring further exploration [9,14,15]. Researchers have found that the active participation of electronic lecturers (e-tutors) is significantly and positively related to the course satisfaction [13,15,16], and active participation plays a crucial role in maintaining attention in online courses [7].

The impact of the COVID-19 pandemic on online learning was drastic and unpredictable. Some studies show this pandemic had an impact on psychological and academic achievement, and lower levels of social-emotional, cognitive, and metacognitive challenges in higher education institutions [17]. Some medical online courses educators had difficulties in teaching clinical skills and assessing learning during COVID-19, which impacted medical education [18]. Emotional impacts, low quality of life, anxiety, and depression are other crucial mental health issues among students and university staff [19]. Not only that, in the secondary level of education, concentration, engagement, ability to learn, and self-worth

from learning were significantly lower, which were also impacted by the pandemic [20]. Learners with disabilities, likewise, are at the highest risk of being left behind when schools close for too long [21]. Despite these impacts and challenges, education must go on no matter what kind of approaches are used, and no matter what conditions. This is a form of lifelong learning and a human right, as acknowledged by the United Nations [21] and UNESCO [22]; hence, when the Covid pandemic struck the world, the only way to sustain education was at a distance: online, radio, and television. Whether we are ready or not, teaching and learning cannot be paused, and there needs to be an effective solution in delivering the content online.

### 2.2. Tools in Online Learning during Crisis

There were several popular online learning tools that educators utilized during the pandemic when conducting online classes. Video conferencing software, such as Zoom, Google Meet, Skype, Webex, Panopto, Echo360, Microsoft Teams, BlueJeans, GoToMeeting, and Join me, etc., became popular. The Zoom application, for example, is the most widely utilized, and is the most preferred video conferencing software, due to its user-friendly design and great user interface [23]. Another popular software is Google Meet, which has the same function. In addition, Microsoft Teams in Office 365 not only has a video conferencing function, but also has a Learning Management System (LMS), which was also a commonly used tool in higher education during the pandemic crisis.

Learning online is more interesting with additional interactive activities. Additional software or apps are strongly recommended, so that online classes can be more engaging, interactive, and self-directed. Kahoot, for instance, is a game-based online quiz embedded with gamification elements to ignite engagement. Quizizz is also an interactive software that enables independent study, as well as the ability to save, stop, and continue at any time. Those are examples of online applications that were user-friendly during the pandemic crisis. With these tools, educators are able to manage their teaching, social, and cognitive presence in a more systematic manner [24].

### 2.3. Community of Inquiry Model (CoI)

The CoI model was developed to measure the ability of an instructor to achieve first-order constructs, such as designing and organizing, facilitating, and directing instruction, to represent the summated score of a second-order concept: teaching presence. This is followed by another second-order concept known as social presence, which denotes online social activities such as emotional expression among peers, open communication, and group rapport as first-order constructs. Similar to the previous concept, the final second-order construct is cognitive presence, which refers to how students reflect on and interact with the content. The combined score is determined by triggering, exploration, and integration resolution [14]. Scoring details are broken down in detail in Table 1.

**Table 1.** Properties of The Measurement Items. Adapted with permission from ref. [25], 2007. D. R. Garrison and J. B. Arbaugh.

| First-Order Constructs | Second-Order Formative Constructs | Definition | No. of Items |
|---|---|---|---|
| Design and Org. | | The development of the process, structure, evaluation, and interaction component of the online courses. | 4 |
| Facilitation | | Establishing and maintaining online discussion through modelling of behaviors, encouragement, supporting, and creating a positive online learning atmosphere. | 6 |
| Direction Instruction | | Describes the instructor's role as a subject matter expert, and sharing knowledge with online learners. | 4 |

**Table 1.** *Cont.*

| First-Order Constructs | Second-Order Formative Constructs | Definition | No. of Items |
|---|---|---|---|
| | Teaching Presence (TP) | The ability of instructors to design, facilitate, and direct cognitive and social processes to produce relevant and meaningful learning outcomes. | 14 |
| Affective Expression | | Emotions/feelings. | 3 |
| Open Communication | | Risk-free expression. | 3 |
| Group Cohesion | | Encouraging collaboration. | 3 |
| | Social Presence (SP) | The ability of learners to project themselves as real people in the community of learners. | 9 |
| Triggering | | An issue, dilemma, or problem (perplexity). | 3 |
| Exploration | | Learners search for information to gain knowledge and make sense of the problem (information exchange). | 3 |
| Integration | | Gain meaning from the ideas developed during the exploration phase (connecting idea). | 3 |
| Resolution | | Applying new knowledge | 3 |
| | Cognitive Presence (CP) | The ability of learners to construct meaning through the continuous reflection and discourse. | 12 |
| Course Satisfaction (CS) | | How much the learners are satisfied with their online course based on course goal, course content, course recommendation, course discussion, and overall course satisfaction. | 4 |

In general, CoI is tailored for the development of theories of engagement, and can model how learners interact and learn online [10,26]. Many researchers have adopted Garrison's research to identify factual evidence on the relationships that exist between teaching presence, social presence, and cognitive presence [14,26–28]. These presences are related to the interaction and the learner's engagement in order to make sure that learning takes place [16]. In other words, engagement and interaction are vital elements in the context of the CoI framework [26].

### 2.4. Student Course Satisfaction and Presences

In this context, satisfaction refers to the quality of the online course being offered. Satisfaction is also a concept that reflects the consequences and mutuality that occur between students and their instructors. As evidenced by researchers [2,29], student satisfaction is vital to the success of an online program, and instructors and institutions must put in effort to meet student needs, and to achieve the goals of the learning environment.

In reality, however, there will be many constraints that distract from online presence. The main issues with online courses are the low level of participation and poor interaction in the virtual environment, which cause dissatisfaction with online courses, as well as cause students to leave them quickly [11,13]. There are many studies about the connection between satisfaction and presence. Several researchers have noticed that the immediacy of teaching presence by an online instructor is very important in online learning [1]. Teaching presence and pedagogical skills are important for student success. Student–instructor communication is repeatedly rated as high in research on online learning [14]. Social presence also contributes to satisfaction [30–32], meaning that there is a connection between social presence and student satisfaction, and it is acknowledged as a significant predictor for learner satisfaction. Social presence is very poor due to learners' lack of familiarity with the system and a number of related factors, such as lack of skill, interest, and motivation [16,31,33].

A meta-analysis study [34] found a moderately large average positive correlation between social presence and satisfaction. Bandura's social cognitive theory, Anderson's interaction equivalency theorem, and Tinto's social integration theory have been reviewed regarding presence in the design of online courses, and they support that social interaction is essential in learning [35]. Additionally, teaching presence is essential to achieving student satisfaction in online courses [29,30]. A lack of immediate feedback from instructors and peers leads to dissatisfaction with the course [36]. Despite the lack of a positive response, unease about working with anonymous peers and expressing opinions in public forum discussions has also led to dissatisfaction. Likewise, other research supports that teaching presence is a very important factor affecting student satisfaction and persistence with online learning [2,30,37].

Nevertheless, the results of other studies are equivocal with regard to student satisfaction and course satisfaction, both of which have their own similar components [38–41], and follow-up research is needed to understand and explain the relationship between teaching presence and satisfaction. Experience and technological comfort [42], the number of assignments [43], and course content [33] were among the aspects influencing presence, which is relevant to online instructors and may add to the findings of others. The Malaysia Critical Agenda Project (CAP) is revising The National E-learning Policy (DePAN1/2) to improve the safety and quality of online teaching, and teaching presence [44,45] may reduce dropout rates [7].

In short, the main recommendation is to further investigate presence and other relevant variables to enrich the literature on the first order of the sub-categories of each second-order presence variable and course satisfaction. As of now, there is a lack of literature and uncertainty about how the presence and satisfaction factors in the CoI model occur in LMS. The quality of research on presence and online learning varies. Although preliminary research suggests presence is related to satisfaction, interaction, and learning, there are still numerous unanswered questions that could lead to interesting research results and additions to the literature.

## 3. Research Model and Hypotheses

In this study, the CoI model was modified to include course satisfaction (Figure 1). The elements in the CoI may have an effect on the effectiveness of the online courses offered, and based on the past studies mentioned earlier, the direction of each of the hypotheses is formulated. Thus, the following hypotheses are worthy of testing:

**H1**. *Teaching presence will be positively related to course satisfaction.*

**H2**. *Social presence will be positively related to course satisfaction.*

**H3**. *Cognitive presence will be positively related to course satisfaction.*

To test the developed model, Partial Least Squares (PLS) will be applied. PLS is a second-generation multivariate technique that analyzes both measurement models (relationships between constructs and their corresponding metrics) and structural models to reduce error variance [46]. According to Hair [47], if there are more than seven variables in the study framework, the model is complex, thus suggesting the PLS is a better application for testing the hypotheses proposed in the study. Moreover, as the model is complex, this study adopted Smart PLS software to analyze the research hypotheses.

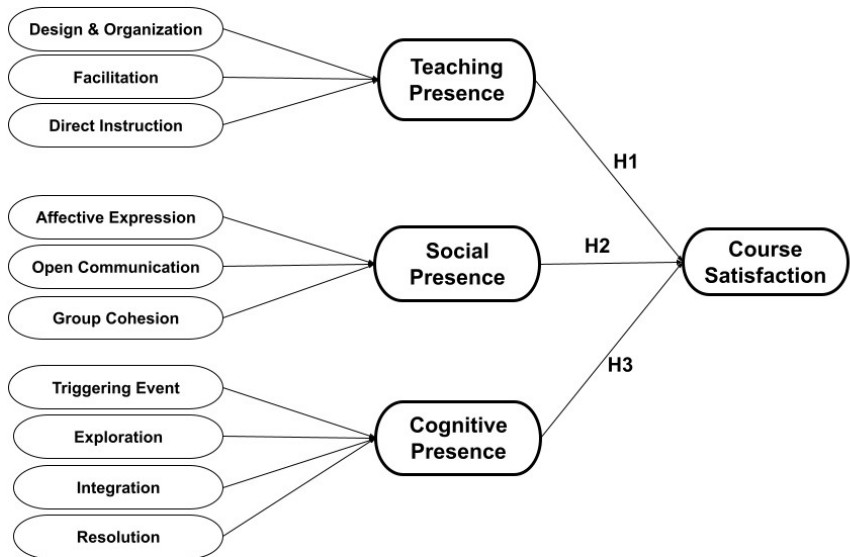

**Figure 1.** The Measurement Model.

## 4. Methodology

### 4.1. Research Design

This cross-sectional survey research design employed a questionnaire to measure the research participants' perceptions on teaching and social and cognitive presence with regard to course satisfaction.

### 4.2. Instrument and Assessment of Goodness of Measure

The CoI instrument was adapted from Garrison [14]. A five-point Likert scale ranging from 1 = strongly disagree to 5 = strongly agree, was used to collect teaching presence, social presence, and cognitive presence as independent variables. The scale was accompanied by course satisfaction items, which were adapted from Artino [38]. Therefore, a six-point Likert scale, ranging from 1 = very dissatisfied to 6 = very satisfied, was used to measure course satisfaction as dependent variable items. Note that the differences in the Likert scales of both variables are to minimize variance issues in the common method while analyzing the data [48].

There are many instruments that measure student satisfaction from different angles. Even though there are studies on satisfaction, in many of them, it was noticed that several items did not specifically focus on a course itself, but on other angles, such as MOOCs, technical issues, services' learning portals (i.e., LMS), Internet quality, etc. Other items combined social presence and teaching presence in the CoI [39–41,49,50]. The authors specifically employed this measure in order to focus on the course.

Table 1 also presents all of the constructs and their meanings, as well as the number of items used in this study. Second-order formative structures, also known as reflex formative type II higher-order models, that include pedagogical presence, social presence, and cognitive presence, have been conceptualized [51]. The repeated indicator approach has been proposed in the PLS literature to model second-order factors in PLS analysis.

### 4.3. Survey Participants

The participants who were involved in this study are first-semester undergraduate students and postgraduate students studying at Universiti Kebangsaan Malaysia (UKM). The number of participants was selected based on a blended learning report on the usage of the university learning portal. The report was gathered for the courses being offered that achieved the minimum requirements for blended courses stated in DePAN: (1) at least seven "Course Materials" files (in any format) uploaded AND a course synopsis; (2) at least four activities/posts; (3) at least two submitted assignments.

Since the study has specific criteria to ensure the validity of the respondents, the purposive sampling method was applied in this study [52]. According to Hair and Ngah [47,53], the minimum sample size was estimated using GPower 3.1.9.4. With three predictors and using the power of 0.8 and the moderate effect size, as proposed by Gefen and Tan [54,55], the study required a minimum sample size of 77. Therefore, the total sample of 422 students is beyond the minimum recommended sample size.

### 4.4. Data Collection and Analysis Procedures

The online structured questionnaires were designed using Google Forms, and the link was emailed to 422 students taking the courses that were determined to fulfil the DePAN requirements. The students came from all of the courses offered in that particular semester. The list of the students' emails was provided by faculty administrators, and students were selected according to a blended learning report. The process of emailing the students took about 2–3 days. Students were invited to participate in the survey. The survey was conducted online over four weeks, with three soft reminders to encourage the participants to take part in the study.

## 5. Data Analysis and Results

### 5.1. Data Analysis

In total, 217 students responded to and completed the online survey and were included in the analysis, resulting in a return rate of 51.65%. To assess the multivariate skewness and kurtosis, the WebPower statistical power analysis online tool (https://webpower.psychstat.org/models/kurtosis/results.php?url=6d33e7abcc18d1b1f1cec513e8427c84, accessed on 18 April 2019) was applied, as suggested by previous researchers [47,56,57]. The results showed that the data obtained in this study did not have a normal multivariate distribution, with Mardia's multivariate skewness of $\beta = 3.30$, $p < 0.001$; and Mardia's kurtosis of $\beta = 26.02$, $p < 0.001$.

SmartPLS was selected as the nonparametric multivariate analysis software for variance-based Structural Equation Modeling (SEM). SmartPLS Version 3.3.2. [58] was used to analyze the data. Following the two-stage approach proposed by experts [52,59], the analysis modelled the measurements and structure of the study. The onset of convergent and discriminant validity precedes the structural model. The bootstrapping method (5000 re-samples) was used to test the hypotheses of the study [60,61].

### 5.2. Common Method Bias

Common method variance (CMV) may arise due to dependent and independent variables calculated from the same participant [48,62]. The study applied the full collinearity analysis proposed by Kock and Mansor [63,64]. All of the variables in the study were regressed to the common variable. Common method bias is considered to be severe for a study if the variation inflation factor (VIF) value $\geq 3.3$. Table 2 shows that all the VIF values were lower than 3.3, thus indicating that the common method variance was not an issue in the study.

**Table 2.** Full collinearity.

| Cognitive Presence | Course Satisfaction | Social Presence | Teaching Presence |
|---|---|---|---|
| 2.215 | 1.588 | 1.659 | 1.997 |

### 5.3. The Measurement Model: Convergent Validity and Discriminant Validity

The first analysis is to establish the convergent validity. The convergent validity will determine which items measure each concept and their agreement with that particular concept [60,65]. The loading and average variance extracted (AVE) must be $\geq 0.5$, whereas the composite reliability (CR) should be $\geq 0.7$ [52,66]. As shown in Table 3, all of the

loadings, AVEs, and CRs of the study exceeded the recommended values, confirming that the study established convergent validity.

**Table 3.** Measurement Model of First-order Constructs (Reflective).

| First Order | Item | Loadings | AVE [1] | CR [2] |
|---|---|---|---|---|
| Design and Organization | TP_DO1 | 0.863 | 0.698 | 0.902 |
| | TP_DO2 | 0.884 | | |
| | TP_DO3 | 0.811 | | |
| | TP_DO4 | 0.779 | | |
| Facilitation | TP_F10 | 0.746 | 0.640 | 0.914 |
| | TP_F5 | 0.730 | | |
| | TP_F6 | 0.814 | | |
| | TP_F7 | 0.833 | | |
| | TP_F8 | 0.822 | | |
| | TP_F9 | 0.847 | | |
| Direction Instruction | TP_DI11 | 0.780 | 0.702 | 0.876 |
| | TP_DI12 | 0.871 | | |
| | TP_DI13 | 0.860 | | |
| Affective Expression | SP_AE14 | 0.780 | 0.603 | 0.820 |
| | SP_AE15 | 0.795 | | |
| | SP_AE16 | 0.755 | | |
| Open Communication | SP_OC17 | 0.867 | 0.773 | 0.911 |
| | SP_OC18 | 0.898 | | |
| | SP_OC19 | 0.872 | | |
| Group Cohesion | SP_GC20 | 0.725 | 0.637 | 0.840 |
| | SP_GC21 | 0.858 | | |
| | SP_GC22 | 0.805 | | |
| Triggering Event | CP_TE23 | 0.827 | 0.646 | 0.845 |
| | CP_TE24 | 0.840 | | |
| | CP_TE25 | 0.740 | | |
| Exploration | CP_E26 | 0.831 | 0.647 | 0.846 |
| | CP_E27 | 0.839 | | |
| | CP_E28 | 0.740 | | |
| Integration | CP_I29 | 0.814 | 0.610 | 0.822 |
| | CP_I30 | 0.854 | | |
| | CP_I31 | 0.661 | | |
| Resolution | CP_R32 | 0.820 | 0.675 | 0.861 |
| | CP_R33 | 0.837 | | |
| | CP_R34 | 0.807 | | |
| Course Satisfaction | CS35 | 0.843 | 0.772 | 0.931 |
| | CS36 | 0.908 | | |
| | CS37 | 0.863 | | |
| | CS38 | 0.899 | | |

[1] Average Variance Extracted. [2] Composite Reliability.

Once convergent validity is confirmed, discriminant validity is assessed using the proposed heterotrait–monotrait (HTMT) assessment [58]. Discriminant validity is established if the HTMT ratio value is less than 0.9 [67]. Table 4 illustrates the HTMT analysis, and indicates that all of the HTMT values were less than 0.9, as mentioned by [67]; thus, discriminant validity was proven not to be an issue in this study.

For the second-order constructs, the study employed the type II higher-order constructs, which are reflective–formative. By employing the three-stage approach promoted by Ngah and Sarstedt [52,68], convergent validity is established if the VIF values and t-values are significant for the weights [60]. Table 5 shows that the VIF values were less than 3.3 [69] and that the t-values were significant (t-value $\geq$ 1.645) for variables representing teaching presence, social presence, and cognitive presence, but not for open communication (OC) in the social presence (SP) category. Thus, as proposed by Hair [47],

the study should rely on the significance of the outer loading. Because all of the variables passed the significance threshold of 1.645, the convergence validity was confirmed to form the higher-order constructs of the study.

**Table 4.** Discriminant Validity of first-order constructs.

| Constructs | 1 | 2 | 3 | 4 | 5 | 6 | 7 | 8 | 9 | 10 | 11 |
|---|---|---|---|---|---|---|---|---|---|---|---|
| CP-I | | | | | | | | | | | |
| CP_E | 0.880 | | | | | | | | | | |
| CP_R | 0.761 | 0.793 | | | | | | | | | |
| CP_TE | 0.717 | 0.773 | 0.485 | | | | | | | | |
| CS | 0.592 | 0.606 | 0.501 | 0.487 | | | | | | | |
| SP_AE | 0.702 | 0.784 | 0.614 | 0.611 | 0.565 | | | | | | |
| SP_GC | 0.517 | 0.545 | 0.425 | 0.472 | 0.469 | 0.853 | | | | | |
| SP_OC | 0.517 | 0.619 | 0.568 | 0.451 | 0.429 | 0.810 | 0.775 | | | | |
| TP_DI | 0.610 | 0.720 | 0.611 | 0.507 | 0.543 | 0.605 | 0.521 | 0.483 | | | |
| TP_DO | 0.505 | 0.631 | 0.468 | 0.518 | 0.527 | 0.495 | 0.428 | 0.462 | 0.740 | | |
| TP_F | 0.689 | 0.729 | 0.559 | 0.534 | 0.572 | 0.587 | 0.473 | 0.492 | 0.810 | 0.837 | |

**Table 5.** Measurement Model of Second-order Constructs (Formative).

| Constructs | Item | Weights | VIF | Outer Weight | Outer Loading |
|---|---|---|---|---|---|
| Cognitive Presence | CP-I | 0.292 | 2.175 | 2.904 | 17.545 |
| | CP_E | 0.320 | 2.388 | 2.287 | 15.488 |
| | CP_R | 0.295 | 1.847 | 1.755 | 10.643 |
| | CP_TE | 0.291 | 1.922 | 1.691 | 7.840 |
| Social Presence | SP_AE | 0.349 | 1.851 | 3.295 | 16.695 |
| | SP_GC | 0.358 | 1.812 | 1.74 | 9.821 |
| | SP_OC | 0.454 | 1.904 | 1.398 | 10.574 |
| Teaching Presence | TP_DI | 0.245 | 1.988 | 2.107 | 12.481 |
| | TP_DO | 0.349 | 2.253 | 1.858 | 15.446 |
| | TP_F | 0.522 | 2.657 | 3.226 | 25.992 |

*5.4. Structural Model*

Once the measurement model was established, the structural model's predictive power; the portion of variance explained by the exogenous variables, represented with the symbol $R^2$; and path coefficients ($\beta$) (beta and significance) were calculated. The bootstrapping technique was applied with a resampling of 5000 to estimate the significance of the path coefficients [60]. Path estimates and t-statistics were calculated for the hypothesized relationship test, as shown in Figure 2.

Prior to further analysis, it is crucial to confirm that the study was free from multicollinearity issues [53]. The VIF values are all lower than 3.3, indicating that the multicollinearity of the study is not severe [69]; thus, the study can be used for hypothesis testing. Based on the three predictors for course satisfaction, the explained variance ($R^2$) was 0.374, indicating that teaching and social and cognitive presence explain 37.4% of the variance in course satisfaction. For hypothesis testing, the hypotheses are supported if the beta value is aligned with the direction of hypothesis, the t-value is $\geq 1.645$, the *p* value is $\leq 0.05$, and there is no zero between the lower level (LL) and upper level (UL) of the confidence interval [64,70]. Based on the analysis, Table 6 shows that all of the hypotheses were supported.

**Table 6.** Hypothesis Testing.

| H | Relationship | Beta | SE | t-Value | *p* Values | LL | UL | VIF | $f^2$ | $R^2$ |
|---|---|---|---|---|---|---|---|---|---|---|
| H1 | TeachingP -> CSatisfaction | 0.287 | 0.073 | 3.951 | 0.001 | 0.166 | 0.404 | 1.853 | 0.071 | 0.374 |
| H2 | SocialP -> CSatisfaction | 0.155 | 0.058 | 2.683 | 0.004 | 0.053 | 0.244 | 1.674 | 0.023 | |
| H3 | CognitiveP -> CSatisfaction | 0.265 | 0.07 | 3.785 | 0.001 | 0.139 | 0.381 | 2.15 | 0.052 | |

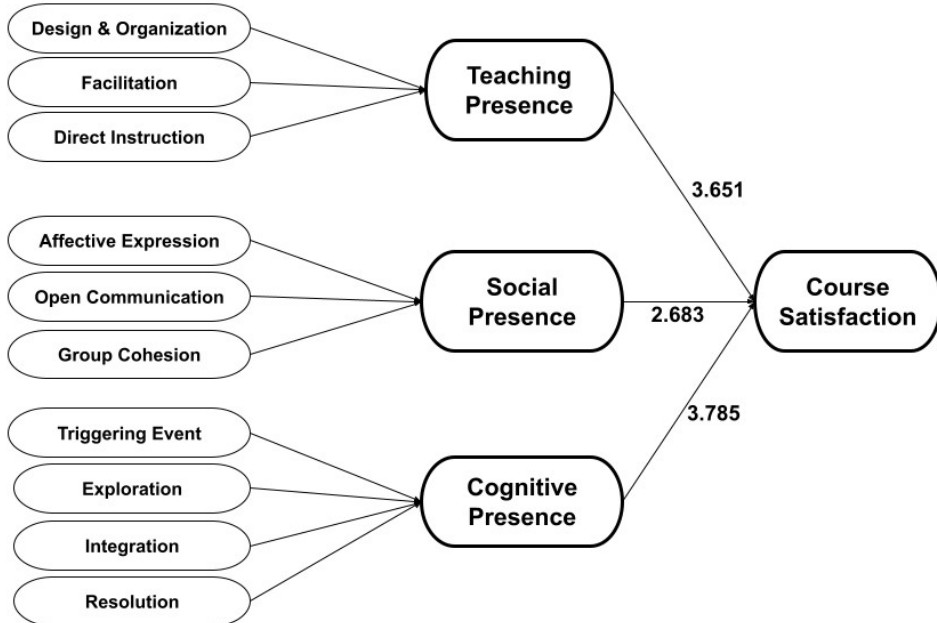

**Figure 2.** Structural Model.

Explicitly, from the analysis, it was found that H1 (teaching presence) was positively related to course satisfaction ($\beta$ = 0.287, *p* < 0.01). H2 (social presence) was positively related to course satisfaction ($\beta$ = 0.155, *p* < 0.01), as was H3 (cognitive presence) ($\beta$ = 0.265, *p* < 0.01). In short, all of the hypotheses in the model were supported. For further analysis, the study also focuses on the effect size of the supported hypotheses. The impact size is divided into low (0.02), medium (0.15), and large (0.35) [71]. Despite the three supported hypotheses, the study found that all of the supported hypotheses have a small effect size, and that the teaching presence has the highest effect size among them, thus indicating that teaching presence is the most important construct of the study. Table 6 summarizes the structural model analysis of this study.

As suggested by Ngah and Shmueli [52,72], PLS makes predictions using a sample-based holdout procedure that generates case-level predictions at the item or construct level via a 10-fold procedure to examine prediction correlations. They suggest that if all of the item differences (PLS-LM) are lower, there is strong predictive power; if the majority is lower, there is moderate predictive power; if there is a minority, there is low predictive power; and if all of them are high, predictive relevance is not confirmed. According to Table 7, all of the errors in the PLS model were lower than those of the LM model, so we can conclude that our model has strong predictive power.

**Table 7.** PLS Predict.

| Item | RMSE | RMSE | PLS-LM | $Q^2$_Predict |
|------|------|------|--------|---------------|
| CS35 | 6.447 | 6.448 | −0.001 | 0.007 |
| CS36 | 0.696 | 0.713 | −0.017 | 0.28 |
| CS37 | 0.748 | 0.772 | −0.024 | 0.25 |
| CS38 | 0.784 | 0.793 | −0.009 | 0.302 |

## 6. Discussion

This study revisits the reliability and the validity of measurements using SmartPLS as a non-parametric analysis software, and variance-based structural equation modelling (SEM) for confirmatory analysis, which is rarely seen in the literature. The results of this study show that the items in the first-order constructs were loaded by more than 0.80. The average variance extracted (AVE) value of all of the constructs was more than 0.70, and the composite reliability (CR) was more than 0.80. Thus, it is shown that the reliability, convergent validity, and discriminant validity of the measurements are supported and confirmed by the analysis conducted according to expert advice [47,72]. This shows that each of the sub-constructs are strongly proven to be good indicators for measuring their respective constructs.

These findings are supported by previous researchers [14] regarding the use of the CoI model in online learning, with previous studies finding that all of the items in the construct are needed to enhance interaction via discussions on online learning. Four items were considered to model course satisfaction as a dependent variable. The first item assessed whether students felt that the course met their goals. The second item assessed students' willingness to recommend the course to people who need to learn the material online. The third item assessed the students' understanding of the course content, and the last item assessed their satisfaction with the online discussion in the course. This construct measures the students' level of satisfaction with the course.

In addition, the main objective of this study was to determine the validity of three hypotheses. The results show that hypotheses H1, H2, and H3 were supported, since a significant relationship with course satisfaction was found between teaching presence, social presence, and cognitive presence, similar to what was found by Cole and Ilduganova [1,30], even though their analyses were different from those carried out in the current study. However, this evidence also confirms Garrison's [14] study, which showed that a higher level of engagement via teaching presence, social presence, and cognitive presence has a positive impact on the level of satisfaction with the course. These significant results show that student satisfaction with online courses is dependent on the characteristics of these presences, and in the Malaysian context, satisfaction is especially dependent on teaching presence. It should be noted that online educators need to enhance their instruction techniques, group communication, and cognitive challenges when conducting online classes.

The results of this study will provide additional guidelines to the Malaysia Critical Agenda Project (CAP) during the revision of The National E-learning Policy to improve the quality of online courses. This will increase the enrolment of online students in the future, especially as public universities in Malaysia are offering Open Distance Learning (ODL) worldwide. As of June 2022, based on Ganbold [73], there is nearly 90% Internet penetration in Malaysia, which is the sixth highest penetration rate across Asia after Brunei, South Korea, Macau, Japan, and Taiwan. However, the Internet penetration rates of Singapore, Thailand, Vietnam, Myanmar, and Indonesia are lower than Malaysia. This scenario indicates that there is more demand for online learning in Malaysia, and once implemented, issues related to dropout can be minimized. The courses will be more sustainable if students are satisfied with the quality of the courses.

Proactive online educators teach effectively because they can better understand the needs of online students in achieving the learning objectives [26]. In this context, online educators need to ensure that they are promoting real instructions, and that their students are experiencing real learning. The construct of course satisfaction in this study can

shed some light on these educational fundamentals. Student satisfaction with the course goals and content demonstrates that students were concerned about the quality of the courses being offered, and how teaching and learning took place. Consequently, increased satisfaction would decrease dropout rates and withdrawal from online courses [6,7]. This suggests that online instructors involved in online teaching could apply various online tools, such as video conferencing software (e.g., Panopto, Echo360, BlueJeans, GoToMeeting, and Join me), supplementary software or apps (e.g., Rapid Refresh, Outgrow, ProProfs, Typeform, etc.), and educational video sites (e.g., Big Think, Brightstrom, CosmoLearning, and MathTV). Online classes should embed fun with additional virtual interactive activities. Thus, the issues of isolation and boredom in learning online could be overcome [6,7], and the learning outcomes could be achieved.

This paper contributes new knowledge to the elements of course satisfaction as they are related to the CoI using reflective–formative type II second-order multivariate statistical analysis. The knowledge is limited in the area. It is also expected that the results of this study will enhance the existing understanding of, and introduce different angles for, managing issues related to online learning. Future research is recommended in order to identify online course satisfaction in other parts of Malaysia. Other recommended research would be comparative studies exploring the practices of online learning in other countries. Another important aspect would be the relationship between CoI to other variables, such as academic achievement and motivation.

In conclusion, online learning has become more significant during the pandemic and post-pandemic periods. This drastic turn could have a long-term impact on education systems. In line with the fourth Sustainable Development Goal (SDG), which emphasizes inclusive and equitable quality education and promotes lifelong learning opportunities for all, the online courses offered in Malaysian higher education institutions are in line with the goals prescribed by UNESCO [22], which aims to market online courses that provide students with qualifications that are recognized worldwide.

**Author Contributions:** Conceptualization, M.K.M.N. and A.H.N.; Data curation, M.K.M.N.; Formal analysis, M.K.M.N.; Funding acquisition, M.K.M.N.; Investigation, M.K.M.N.; Methodology, M.K.M.N.; Project administration, M.K.M.N.; Resources, M.K.M.N.; Software, M.K.M.N. and A.H.N.; Supervision, M.K.M.N.; Validation, M.K.M.N.; Visualization, M.K.M.N.; Writing—original draft, M.K.M.N.; Writing—review & editing, M.K.M.N. and A.H.N. All authors have read and agreed to the published version of the manuscript.

**Funding:** This research was funded by the Faculty of Education's research grant via University Kebangsaan Malaysia (UKM) under the project code GG-2021-007.

**Institutional Review Board Statement:** Ethical review and approval were waived for this study, due to the research presenting no more than minimal risk of harm to participants approved by the Ethics Committee of Faculty of Education, UKM.

**Informed Consent Statement:** Informed consent was obtained from all subjects involved in the study.

**Data Availability Statement:** Original data are not publicly available due to ethical restrictions on identifying participants.

**Acknowledgments:** The authors thank the Faculty of Education, Universiti Kebangsaan Malaysia (UKM) for the opportunity to conduct this research.

**Conflicts of Interest:** The authors declare no conflict of interest.

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
