# Peer review of "The Sustainability of a Community of Inquiry in Online Course Satisfaction in Virtual Learning Environments in Higher Education"

_sustainability, doi:10.3390/su14159633_

Round 1
Reviewer 1 Report
This is an interesting study and the author has collected many statistics to argue the hypotheses. The paper is generally well written. However, in my opinion the paper has some shortcomings in regards to some data analyses and discussion, and I feel the author gives the detailed information of research method but not enough analyses, in particular with the last two tables [6,7]. In the part of discussion, the author should give more explanations about research result, such as CoI model, its relation with students’ satisfaction, and the author’s suggestions or prediction about how to improve students’ satisfaction according to the result. This research should give a full discussion about its result and explain the relation among those methodologies.
In addition, the research meaning should be highlighted in the final part. The research question maybe not so controversial, and it needs creative findings and suggestions in the final part.
Finally, there are many quotations in this research, so the originality is my concern.
Reviewer 2 Report
This paper aims to evaluate learner satisfaction using the community of inquiry as its evaluation framework. Various statistical methods were applied to ensure validity and to test the hypothesis via some form of latent variable modeling.
- In several areas, the sentences do not flow well (e.g., the sentence is incomplete, hard to identify the subject and the predicate, a sentence does not build up the previous sentence, etc.). This makes the work challenging to read. You can see for example the first two sentences already have some grammar misses: "Online learning growth rapidly worldwide due to the market demand and students need. The need normally coming from people who has limitation to go to an institution and meet their teacher or lecturer physically face-to-face especially during Covid-19 condition." As these seemingly minor problems accumulate, the readability of the manuscript suffers.
- The choices of measurements are either not very well-motivated, or the writing prevented the motivation to be obvious to the reader. In fact, in a lot of places, something to the effect of "XX said this, so we did as well." While it could be a valid justification, it is weak.
- There are several existing works investigating the community of inquiry in online learning. I guess the crux of this work is its uniqueness as it is not done yet in the Malaysian context. Why did this warrant an investigation? Are the online learning practices vastly different from those in other countries? After all the analysis is done, what did we learn new?
- Question: How did the course satisfaction survey eventually look like? In Table 3, this looks like reduced to just 4 items whereas there are four different sources. Did you take just one question per source, or did you somehow aggregate the scores for each score? If the latter, how?
- Question 2: What do the abbreviations in Table 2 mean?
- For Table 7, why was CS35 excluded?
- Figure 1 (and most other figures) could be enlarged. A guideline that can be used is checking if the text within the figures is more or less the same size as the article text.
- Table 3 goes beyond the margin (overlaps with line numbers)
- It will be much easier if the table mentioned in the text is displayed immediately after. For example, in this manuscript, Table 1 was mentioned in Line 85 on page 2, but Table 1 itself does not appear until Line 216 on page 6
- Statements: I did not see the data availability and ethics statements
Reviewer 3 Report
Dear authors
Thank you so much for your efforts and a great job in writing your manuscript and preparing it to be published in Sustaiability journal. I have some comments and recommendation and would like to take into consideration when submit your manuscript to be suitable for publication in the Sustainability journal.
Abstract
I recommend to write the abstract it precisely to reflect the actual research you have done. Write one sentence or two sentences and then go to the purpose, research design, instrument,... and the major findings as well as the limitations and future research.
Introduction
It is too short and not included the research problem clearly as well as the contribution of your research to international readers. I recommend to add more and cite new related references. Please update the references, your have references for more than 6 years.
Literature review is week and you need to rename some subheading such as issues in online learning I suggest to rechange it to be challenges, current practice of online learning. Since you are talking about COVID-19, I suggest to add a section about tools in online learning during crisis.
What is the framework of your research?
You did not clarify the relationship between technology acceptance models and Community of Inquiry Model (CoI). There is something missing. Please write about the connection between them.
You used many big concepts such as engagement theory without any citation or definition, I need to read more about it and how it is connected to your research. Do you want to use it as theoretical framework of your study? Please give more details. Do not expect the reader knows everything about your research and the terminology .
Research model and hypotheses: how did you develop the hypotheses?
Methodology
Before talking about the survey participants, please write about your research design, instrument, how did you develop the survey? Did you use existed one? why? or did you develop a new one, how and why?
Data collection and analysis procedures were missing before understanding the data analysis procedures, it is difficult to accept the findings, Please add more about data analysis.
Discussion should be deeply and connect more with previous studies. Please rewrite it
Where is the limitations, theoretical and practical implications of your study and future research. Please add a new section to include the above implications with citation from literature.
Ethical concerns
Please provide relevant information about the Institutional Review Board Statement from the university or the institutions and mention the informed consent from the participants in the study.
Round 2
Reviewer 1 Report
Tables and figures are of lower quality. The former is always broken and the latter cannot be clearly detected. There are also language problems throughout the manuscript.
Author Response
"Please see the attachment."

Reviewer 2 Report
Thanks for your revisions! I would have appreciated it if the response to reviewers included the details on how the comment was addressed (e.g., "In Line XX, we added discussion about aaa citing bbb"). I realize you may have made changes based on my comments, but it was very challenging to ascertain if what you did is what I think it is.
I commented previously that the writing needs some review. I still had some struggles reading through. A combination of free or built-in grammar check tools (e.g., Grammarly) and a style check tool (e.g., Hemmingway App) might be helpful. Reading aloud also helps. In some areas, it feels like it is never-ending (e.g., second paragraph of the introduction), while in some places, it feels abruptly cut (e.g., section 2.1).
I noticed that you considered my comment about the motivation behind the measurements and the uniqueness of the research to be a single item. I believe these are different issues and the response and the edits are less than satisfying. For the motivation behind the measurements, why did you select these measures? Looks like the CoI was explained, but how about the satisfaction survey? As for the uniqueness, I recognize that Malaysia might have problems that can be uncovered by this research, but won't other research results be able to? Why would online learning be different in Malaysia versus other countries?
The figures somehow appear blurry. If you are working with PowerPoint and Word for writing this manuscript, I suggest you try to save the PowerPoint slide as an SVG and paste the SVG to Word.
Finally, which I missed asking previously, where is the sustainability angle here? What is the finite resource that you are trying to optimize for?
Author Response
"Please see the attachment."

Reviewer 3 Report
Authors did a great job in revising the manuscript and addressed my previous comments and feedback. I recommend to accept after revising English language because a minor errors in language structure.
Author Response
"Please see the attachment."

Round 3
Reviewer 1 Report
I think the references must updated because there are excellent studies excluded.
I recommend the following two citations to further explain CoI in online and virtual learning contexts respectively:Yu, Z., & Li, M. (2022). A bibliometric analysis of Community of Inquiry in online learning contexts over twenty-five years. Education and Information Technologies. https://doi.org/10.1007/s10639-022-11081-w
Yu, Z., & Xu, W. (2022). A meta-analysis and systematic review of the effect of virtual reality technology on users’ learning outcomes. Computer Applications in Engineering Education, 1-15. https://doi.org/10.1002/cae.22532
Reviewer 2 Report
Thanks for the revisions. It is mostly easier to read this time. Here are some remaining clarity concerns:
- Line 50: I was not able to follow the connection with international students.
- Line 53: No need for "Thus,"
- Other things in this aspect can possibly be better commented on by a professional copy-editor than me.
In response to your responses:
- Line 78, do you have more recent data?
- Line 208, your comments here are unsupported though. Maybe give a few examples? This also applies to line 214.
- Consider moving items in line 214 earlier. The motivation and uniqueness of the study ideally should be revealed early in the manuscript, possibly in the introduction.
- Line 391 is great!
- Line 427 is very timely and is a great addition.
